# Effectiveness of graded motor imagery protocol in phantom limb pain in amputed patient: Protocol of a randomized clinical trial

**Sandra Rierola-Fochs**, **Jose Antonio Merchán-Baeza***, **Eduard Minobes-Molina**

Research Group on Methodology, Methods, Models and Outcomes of Health and Social Sciences (M₃O), Faculty of Health Sciences and Welfare, Centre for Health and Social Care Research (CESS), University of Vic-Central University of Catalonia (UVic-UCC), Vic, Spain

* josan.merchan@uvic.cat

**Data Availability Statement:** No datasets were generated or analysed during the current study. All

## Abstract

### Objective

The aim of this study is to analyse the effectiveness of the Graded Motor Imagery (GraMI) protocol in phantom limb pain in amputee patients.

### Materials and methods

A randomised clinical trial will be conducted, with two parallel groups and simple blinding, and a phenomenological study with semi-structured interviews. People over the age of 18, with amputation of one limb, with a minimum score of 3 on the visual analogue scale of pain, who are pharmacologically stable and have been discharged from hospital, will be recruited. An initial assessment, a post-intervention assessment (9 weeks) and a follow-up assessment (12 weeks post-intervention) will be performed, in which pain, quality of life, functionality and psychological aspects will be assessed. The aim of the qualitative study is to find out about the experience of living with phantom limb pain and to identify the satisfaction with the intervention. A descriptive, univariate and bivariate quantitative statistical analysis will be performed using the SPSS program, with a 95% confidence level and a statistical significance level of $p < 0.05$. The qualitative analysis will be carried out using the Atlas.ti 8.0 program, where the different interviews will be analysed, coded and categorised.

### Discussion

The GraMI protocol allows the patient to work on motor learning through brain reorganisation, analytical movements, sensory stimulation, and functional activities. In addition, it can help to standardise the use of graded motor imagery in future studies and in clinical practice with this patient profile.

### Trial registration

NCT05083611.

relevant data from this study will be made available upon study completion.

**Funding:** The author(s) received no specific funding for this work.

**Competing interests:** The authors have declared that no competing interests exist.

# Background

Phantom limb pain (PLP) can be defined as discomfort or pain in a missing part of the limb [1]. It is not a syndrome common to all amputees, but it is extremely prevalent. A study in 2005 reported that 60–80% of amputees suffer from PLP [2], and in a subsequent study in 2020 this was reported as 67–87% of amputees [3], although not with the same intensity or frequency of symptoms [4]. PLP after amputation occurs due to multifactorial mechanisms [5]. There are various predictive factors that influence the perception of pain, such as pre-amputation pain, location, sex and time of evolution, among others [5]. In 72% of cases, it appears in the week following the amputation [6]. However, in some cases it can evolve over the course of weeks, last over time and become a source of chronic pain [7]. The persistence of this pain over time may be related to the appearance of psychological disorders, such as anxiety and depression, or alterations in the activities of daily living and in the quality of life of people affected [8].

The root cause of phantom limb pain is not well understood; it may be due to an irritation of nerve endings, a central remapping of sensations or the mismatch of motor commands and visual feedback, which is then interpreted as pain [9]. Clinical findings suggest that both peripheral and central nervous system mechanisms can contribute to the genesis of PLP [10]. Since the late twentieth century, the dominant theory about the origin of PLP is that it is related to the maladaptive plasticity that occurs in the brain after an amputation [11]. This theory argues that adjacent cortical areas can occupy areas corresponding to the amputated limb, causing activation of the affected areas by stimuli from healthy areas [12]. In addition, after an amputation, the representation of this part of the body at the central level may remain intact, but there may be a mismatch between the visual feedback it receives and the perception of that limb in response to pain [13, 14].

More than two dozen techniques can be found in the literature for the management of PLP [5]. However, there is no broad consensus on the best and most effective option [1]. One of the rehabilitation techniques for PLP is graded motor imagery (GMI) [15]. The GMI is a comprehensive rehabilitation programme, consisting of three different techniques (laterality recognition, motor imagery and mirror therapy), applied in three progressive and consecutive stages [16].

GMI was developed in the 2010s and is based on a number of neuroscientific foundations, including neuroplasticity, motor learning, and the use of mirror neurons present in the brain [16]. Mirror neurons are a group of neurons located in the lower parietal lobe, the ventral and dorsal premotor cortex, and the primary motor area of the frontal lobe. These neurons are activated if the person makes a movement themselves, imagines making a movement, or observes a movement by a third person [17]. The progression of the three techniques in the GMI is related to the brain activation caused by each of them. The greatest brain activation is observed when the person performs the movement in the first person, then when they observe it through mirror therapy, then when they perform the explicit motor imagery and finally when they perform the recognition of the laterality or implicit motor imagery. Therefore, in order to achieve brain reorganisation without causing the pain, gradual progress is made [18].

The first technique is laterality recognition or implicit motor imagery, which consists of differentiating the laterality (right and left) of the two hemicossos through images that are presented to the person, has great importance in the planning and reorganisation of voluntary movement [19]. When an image is observed, the person initially assumes which side it is on, then, in order to corroborate the hypothesis, a mental image of the position is made. When this image is conscious, it is called the explicit motor image and activates the primary motor area at the brain level. As a result, the person may notice an increase in pain intensity at the

beginning of the procedure. However, with practice, the person is no longer aware of this mental simulation to corroborate the hypothesis, and goes on to perform an unconscious simulation called an implicit motor image. This activates the premotor cortex, which is connected to the primary motor area but fails to activate it [18].

The second technique is motor imagery or explicit motor imagery, which consists of imagining functional movements or actions without performing voluntary contraction; a mental simulation [16, 20, 21]. The person looks at different positions on an image and has to imagine that they are bringing their limb to the position on the image and returning it to the starting position. These first two techniques aim to reorganise the representation and brain pattern through neuroplasticity, activating the premotor areas and supplementary motor areas, and desensitising the primary areas [20, 22].

The third technique is mirror therapy, which involves placing a mirror in the sagittal plane between the limbs, so that the amputated limb is behind the mirror. The person is then asked to observe the reflection of the healthy limb in the mirror [23]. This aims to teach the brain that there can be pain-free movement and to pre-activate injured areas to reduce malplasticity after amputation [24].

There is currently little evidence on the implementation of GMI in PLP specifically [16, 25], as there is no standardised way to use it in this patient profile. As a result, a systematic review has been conducted prior to this study to determine the effectiveness of GMI and its components on PLP in the amputee patient. Through this review, it was possible to define a protocol that has been validated by a group of experts in the field of neurorehabilitation and/or pain via a study using Delphi methodology [26], which has resulted in the GMI-based GraMI protocol.

The GraMI protocol consists of three progressive techniques and can be performed by the patient autonomously and individually at home, with follow-up by a professional. It is necessary for the patient to use a mobile application and have access to a mirror box in order to carry out the intervention. Currently, the majority of the population has access to digital technologies, which generate more motivation, dynamism, adherence and continuity with the intervention compared to other techniques [27]. Therefore, a mobile application will be designed in advance for this intervention.

The aim of the present study is to analyse the effectiveness of the GraMI protocol on PLP, quality of life, functionality, and associated psychological aspects in the amputee patient. In addition, it also aims to understand the experience of people with PLP and assess their satisfaction with the intervention. Our hypothesis is that the GraMI protocol can decrease PLP and as a result improve quality of life and functionality, and decrease the associated psychological factors.

## Materials and methods

### Study design and setting

A randomised clinical trial will be performed, with two parallel arms and simple blinding, following the recommendations of the Consolidate Standards of Reporting Trials (CONSORT) [28], alongside a phenomenological study with the participants of the intervention group, through semi-structured interviews, following the recommendations of the Standards for Reporting Qualitative Research [29]. In addition, the SPIRIT guidelines [30] have been followed. Fig 1 outlines when each study component occurs. The study was registered with ClinicalTrials.gov (registration number: NCT05083611). The clinical trial will be carried out at the community level, where the intervention will be carried out by the participant themself autonomously and individually, with supervision and follow-up by the main researcher of the study.

| STUDY PERIOD | | | | | |
|---|---|---|---|---|---|
| | **Enrolment** | **Allocation** | **Post-allocation** | | **Close-out** |
| **TIMEPOINT** | **-t1** | **0** | **t1 (baseline)** | **t2 (Post-intervention)** | **t3 (3 Months)** |
| **ENROLMENT:** | | | | | |
| Eligibility screen | X | | | | |
| Informed consent | X | | | | |
| Allocation | | X | | | |
| **INTERVENTIONS:** | | | | | |
| [*Routine of physiotherapy*] | | | | | |
| [*GraMI protocol*] | | | | | |
| **ASSESSMENTS:** | | | | | |
| SF-MPQ | | | x | x | x |
| EQ-5D-5L | | | x | x | x |
| FIM | | | x | x | X |
| Beck | | | x | x | x |

**Fig 1. Outlines when each study component occurs in accordance with the SPIRIT guidelines.** SF-MPQ: Short form McGill Pain Questionnaire, EQ-5D-5L: EuroQool-5D-5l, FIM: Functional Independence Measure, Beck: Inventario de depresión de Beck, t: Timepoint, GraMI: Graded motor imagery.

## Ethical considerations

The protocol of this clinical trial was approved by the Ethics Committee of the University of Vic-Universitat Central de Catalunya (UVic-UCC) with registration number 185/2021, by the Ethics Committee Quirón Salud-Catalunya with registration number 2022/04-COT-ASE-PEYO and the Ethics Committee FORES with registration number 2022188/PR319 following the criteria required by the Helsinki Declaration, as well as the Organic Law 3/2018 (December 5) on the Protection of Personal Data and Guarantee of Digital Rights. During the first contact with the hospitals, the project will be explained orally and those who agree to participate will be provided with the information sheet via email. In addition, prior to the start of the study, informed consent will be sought from participants in writing and signs, they will be provided with the information sheet and the contact details of the principal investigator to answer any questions or make clarifications, and they will be informed on the disposition of the waiver sheet in case they want to leave the study at any time.

## Sample size

The sample size was calculated using the main pain variable using the G * Power 3.1 program, taking into account an alpha error of 0.5, a beta error of 0.8 and an effect size of 0.81 [31]. This

**Table 1. Trial inclusion, exclusion and elimination criteria.**

| Inclusion Criteria | Exclusion Criteria | Elimination Criteria |
|---|---|---|
| 1. People over the age of 18.<br>2. Amputation of one limb, upper or lower.<br>3. Presence of phantom limb pain.<br>4. A minimum score of 3 on the analogue visual pain scale.<br>5. Under pharmacological treatment for pain<br>6. Pharmacologically stable<br>7. Have been discharged from hospital | 1. Severe visual impairment (hemianopsia).<br>2. Neurological, cognitive impairment (attention deficits).<br>3. Previously received treatment with GMI. | 1. Changes in medication during the procedure that directly affect the main variable pain.<br>2. Changes in the use of the prosthesis |

effect size has been extracted from a study that evaluates PLP using the McGill Pain Short-Form Questionnaire and a mirror neuron intervention. The total number of participants will be 50, with 25 in each group. A non-probabilistic sampling rate type will be used.

## Participants

Participants will be recruited through the reference health professionals of different hospitals and centres that have amputee units, including the Hospital Universitari de Vic, Hospital de la Santa Creu in Vic, the Hospital de Sant Jaume in Manlleu, MC Mutual and the Asepeyo of Sant Cugat del Vallés, Catalonia (Spain), which will assess their eligibility together with the principal investigator (SRF) according to the established inclusion criteria. The principal investigator (SRF) will then contact participants by telephone to offer them the opportunity to participate and send them an online fact sheet with the details of the study. Those who agree to participate must sign the informed consent form and be informed that they have a waiver form so that they can withdraw from the study at any time if they wish. Inclusion and exclusion criteria for study participation are detailed in Table 1.

## Randomisation and blinding

During the recruitment, an external person who will not participate in the intervention, analysis or interpretation of the results will randomise the participants using sealed envelopes that will assign them to the control or intervention group. Due to the nature of the intervention, and given that the characteristics of the intervention are explained in the intervention sheet, we cannot guarantee that participants will not deduce which group they belong to. Additionally, the therapists who will perform the intervention cannot be blinded either.

## Procedure

The present study consists of three different consecutive stages.

## Stage 1: Design of the application

This stage will consist of the design and development of a mobile application by a team of UVic-UCC engineers, with the support of the research team. The estimated design time will be 6 months, which will include piloting to test its usability and accessibility. This application will allow the patient to carry out the first two techniques of the GraMI protocol, laterality recognition and motor imagery. During its design, a gamified application will be developed to encourage the motivation and adherence of the participant, and aspects will be defined that will help the principal investigator in the supervision and follow-up of the intervention. Each activity will be recorded numerically and graphically in order to facilitate visualisation and encourage improvement.

## Stage 2: ECA

The RCT has been designed following the *Template for Intervention Description and Replication (TiDier)* [32] guide in order to adequately describe all aspects related to the intervention and to encourage the replication of the study.

The duration of the intervention, both in the control group and in the intervention group, will be 9 weeks plus 12 weeks of follow-up. Both groups will initially receive an educational session, in person or online, where the physiology of PLP, brain plasticity and the importance of rehabilitation will be explained. The session will last approximately 30 minutes, where it will be explained in detail that after an amputation there is an incongruity of visual, sensory and motor information and maladaptive plasticity due to adjacent areas occupying the areas injured by the amputation. This educational session will be given by the principal investigator of the project, who has clinical, teaching and research experience in the field of neurorehabilitation and pain.

**Intervention in the experimental group.**    The participants from the experimental group will perform the GraMI protocol [26]. This consists of three consecutive and progressive techniques: laterality recognition, motor imagery and mirror therapy. Each participant will perform the intervention independently at home, but will be supervised and guided by the principal investigator. At the beginning of each of the three techniques, the principal investigator will perform an educational session of 30 minutes, explaining the procedure of the phase and the objectives to be achieved, as well as answering questions and clarifications. The explanation of the intervention is divided into three educational sessions to facilitate the interpretation by the participant and thus encourage continuity and follow-up.

*1. Laterality recognition*. This will be performed for 10 minutes, twice a day, on 5 consecutive days per week (resting over the weekend), over 3 weeks. Images of the lower or upper extremities will appear via the GraMI mobile app, depending on the level of the injury, and the participant must identify as soon as possible which side they belong to (right or left). The 3 weeks of intervention will be divided into three progressive phases with ascending difficulty; for the first 5 days, images will appear in the neutral position; then for the next 5 days, images will appear in different planes in space; and for the last 5 days, images of functional activities and interaction with objects will appear. The accuracy of each activity and the time used to identify each image will be recorded by the application. The progress is shown in Fig 2.

*2. Motor imagery*. This will be performed for 10 minutes, twice a day, on 5 consecutive days per week (resting over the weekend), over 3 weeks. This technique will also be carried out using the mobile application. In this case, images of the lower or upper extremities will appear, depending on the level of the injury, and the participant will have to imagine that they are placing their limb in the same position as the image and returning it to the initial position, without performing voluntary contraction, undergoing a mental simulation alone. The same progression of images and days as the previous phase will take place over the 3 weeks.

Before starting the motor imagery phase, the person's imagination will be assessed through the kinaesthetic and visual imagination questionnaire (KVIQ) [33]. This is a 10-item questionnaire in which the physiotherapist teaches different movements and the person must rate their ability to imagine them in the first person. In the event of an impairment of imagination, the person will be taught different strategies to do so.

One alternative strategy is to show the participant videos and images of the affected limb before the intervention in order to pre-activate the appropriate areas of the brain through the observation of the action. Another strategy is to try to find a factor or situation that helps the participant to imagine it. However, pain can occur during the imagination phase. If pain appears, the person can be instructed to first imagine the movement with the healthy limb. If

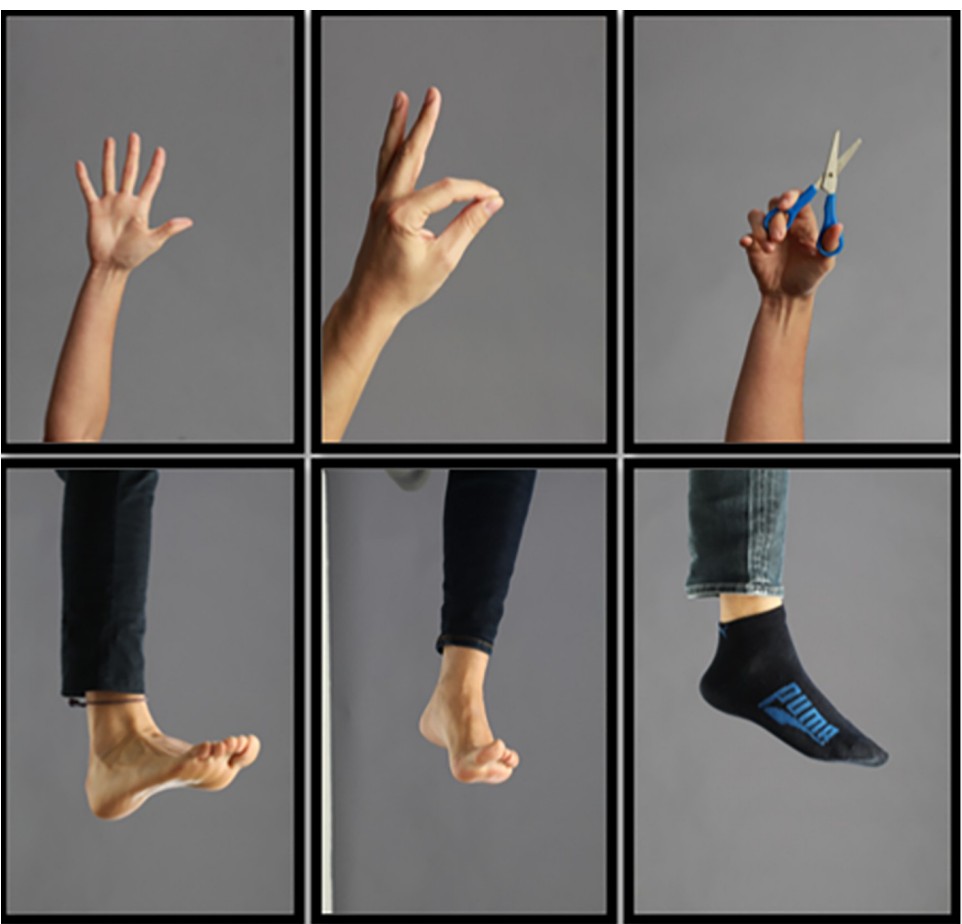

**Fig 2. Progression of the images during the three weeks in the phase of laterality recognition and motor imagery.**
First week: Images in neutral positions in the different planes (left), second week: Images in different positions in the different planes (centre), third week: Images of functional activities or with object interaction (right).

the pain is still triggered, we can ask them to imagine the movement in the third person. These strategies will be modified throughout the intervention to allow the person to realise the mental imagination in the first person.

*3. Mirror therapy*. Finally, the third technique will be performed with the mirror box for 20 minutes, once a day, on 5 consecutive days per week (resting over the weekend), over 3 weeks. This technique will also be divided into three progressive phases with ascending difficulty: for the first 5 days, the participant will perform analytical movements of the joints involved in the amputation; for the next 5 days, they will perform sensory stimulation (texture, pressure, temperature, etc.); and for the last 5 days, they will perform functional activities and interaction with objects. The progress is shown in Fig 3. At the beginning of the intervention, the person will be asked not to try to perform the movement with the affected limb, otherwise the incongruity of information can be enhanced. When the person gets used to the intervention, they will be asked to make the movement with the stump in order to create the most real activation possible in the brain [11, 15, 23].

In order to follow up the intervention, the main researcher of the study will contact the participant by telephone once a week in order to resolve possible doubts, make clarifications or simply to remind them of the importance of follow-up of the intervention. In addition, the

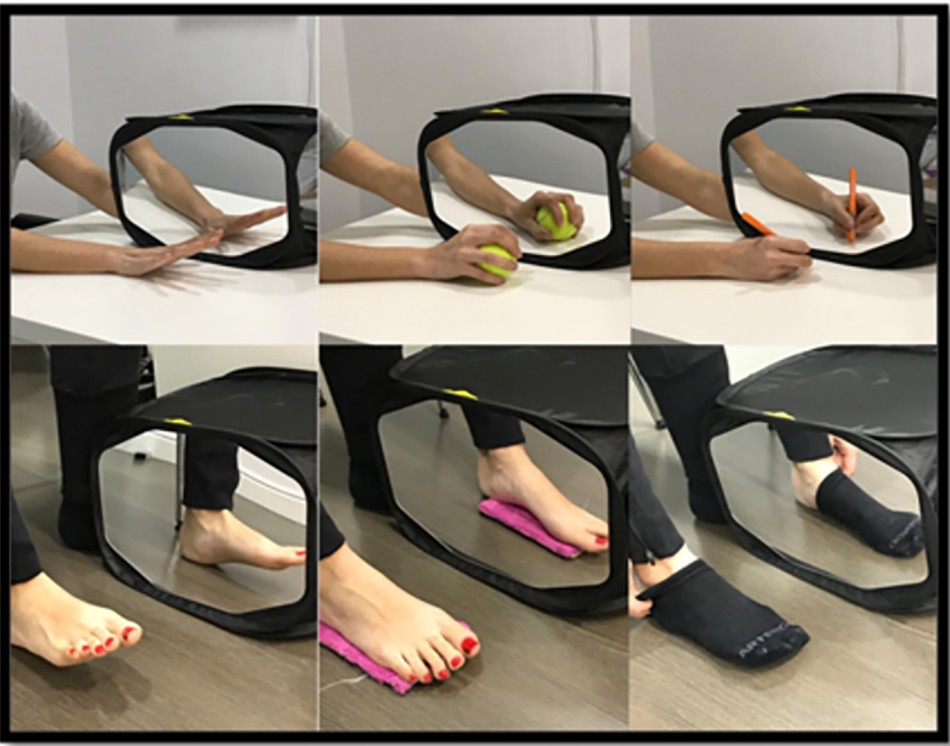

**Fig 3. Progression of the images during the three weeks in the phase of mirror therapy.** First week: Analytical movements of the joints involved in the amputation (left), second week: Sensory stimulation with different temperatures, textures, densities and/or vibration (centre), third week: Interaction between objects or functional activities (right).

application will generate a daily notification in order to encourage continuity with the intervention.

**Control group intervention.** In addition to the educational session, the control group will continue with the conventional treatment they are receiving at that time, plus the pharmacological treatment. If at that time they are not undergoing any conventional treatment, they will only recive the pharmacological treatment. The main line of treatment in the clinical setting for phantom limb pain is pharmacological. In addition, they will be given a diary and they will be advised to record their activities, specifying the nature, frequency and duration of each activity in order to have a record of whether the activities carried out can influence PLP. At the end of the study, if the hypothesis that the GraMI protocol is effective in reducing PLP is verified, participants in the control group will have the option of undergoing the intervention.

## Stage 3: Phenomenological study

A phenomenological study will be carried out, as it allows us to describe a person's experience through semi-structured individual interviews [34]. During the study procedure, the objectives and methodology of the qualitative study will be explained in person or by telephone to those people who are part of the intervention group. Those who agree to participate will be sent an email with the informed consent form and the information sheet. The appropriate sample size is 15 participants, taking into account the purpose of the research and analysis, and the time and resources available [35]. However, saturation of the data will indicate the true sample size that is needed [36]. Once the intervention is over, each person will be allowed to choose the

time and format (online or face-to-face) for which the interview is best for them. The interview will be semi-structured and based on a script, but will be adapted as the interview progresses. It will be structured in three large blocks; an initial introductory block on demographic details, a second block to learn about the experience of suffering from DMF, and a third block to assess satisfaction with the intervention received. The approximate time for each interview will be 30–45 minutes.

The researcher conducting the interview will be an external person who has not conducted the intervention or assessments, so as not to influence the questions and answers. This person has theoretical sensitivity about PLP and mirror therapy and this will make it easier to delve deeper into the topics during the interviews and better understand the experience. Two members of the research group will then transcribe and analyse the interviews, thus increasing the reliability and objectivity of the analysis and results. The interviews will be recorded in voice and video format in order to enable the later transcription to be as good as possible, and allowing analysis of the facial expressions and body language of the participant. Only the research group and the researcher who conducted the interviews and transcripts will have access to the recorded material. Once the transcripts have been completed, they will be sent to each participant to corroborate the information. This process will also be performed after the analysis of the results. At the end of the study, the recordings will be deleted.

## Variables

To ensure blinding during the assessment, two external researchers will perform the assessments. These researchers will be two physiotherapists with clinical experience in the field of neurorehabilitation and/or pain, trained in the research methods, and calibrated with each other to minimise bias. They will be blinded to group assignment. There will be an initial assessment, a post-intervention (9-week) assessment and a follow-up (12 weeks post-intervention). The assessments will be carried out in person at the homes of the participants and will last approximately 25–30 minutes. Assessments will occur in the same space and at similar times to avoid factors that could influence the results. Socio-demographic variables (age, gender, marital status, background) and characteristics of amputation (side, level, cause, time of evolution) will only be included in the initial assessment. The dependent variables will be pain, quality of life, functionality and associated psychological aspects, which will be collected throughout the three assessments and always in the same order.

**Main variable.** Pain is the main variable and will be assessed through the *McGill Pain Short Form Questionnaire* [37]. This is a self-administered questionnaire that assesses pain from a quantitative and qualitative point of view. It consists of 15 pain descriptors, of which 11 are sensory categories and four affective. Each descriptor contains three columns, which are categorised as 1: medium, 2: moderate and 3: severe. The participant must mark the degree to which that descriptor applies for their pain; if it is not present, they can leave it blank, indicating a value of 0. In addition, the questionnaire contains a visual analogue scale of pain, which is a reliable and validated metric for pain assessment [38]. There is no cut-off point; the higher the score, the greater the pain [39]. This questionnaire takes between 2 and 5 minutes to answer.

**Secondary variables.** Quality of life will be assessed using the *EuroQol-5D-5L scale* [40]. This is also a self-administered scale, consisting of five dimensions (mobility, self-care, regular activities, pain/discomfort, and anxiety/depression). Each dimension is made up of five score levels, indicating no problems, mild, moderate, severe and extreme problems. The participant is asked to indicate their state of health by ticking the box indicating the most appropriate statement in each of the five dimensions. In addition, it contains a numerical scale from 0 to

100 to quantify the degree of health on the day of the assessment. It takes between 5 and 7 minutes to answer.

Functionality will be assessed using the *Functional Independence Measure* [41], which consists of 18 items, divided into seven levels, which are personal care, sphincter control, mobility, transfers, walking, communication and social cognition, which assess the motor and cognitive ability of the person. Each item is scored on an ordinal scale of 1 to 7 points. The range of scores ranges from 18 to 136; a greater score indicates more independence on the part of the patient to perform the task associated with each item. It takes 5 minutes to answer.

Psychological aspects will be assessed using the *Beck Depression Inventory-II* self-administered scale [42]. This scale allows the detection and assessment of the severity of depression. It consists of 21 questions. Each question is quantified on a 4-point scale ranging from 0 to 3, with 0 indicating that the participant has been experiencing no symptoms and 3 that they have been experiencing symptoms severely over the past 2 weeks. The scores range from 0 to 63 points. A higher score indicates a greater severity of depressive symptoms. It takes between 5 and 10 minutes to answer.

The participants' medication will be collected from 2 weeks before the start of the intervention until the end of the study. The drugs used and the prescribed doses will be collected. These data will be grouped together with the sociodemographic variables and the characteristics of the amputation during the first assessment. Any changes that occur during the intervention will be monitored. In the event that they occur, they will be noted and analysed, as these changes may affect the main variable (pain). As mentioned above, one of the inclusion criteria is that participants are pharmacologically stable, as changes in medication could influence the results of the assessments.

## Adverse events

The risk of participating in this study is very low, as it is not an invasive intervention. If the assigned intervention results in a worsening of the participant's symptoms or they request to stop the intervention, the participant will be removed from the trial without consequences. In the event that the hypothesis is verified and the intervention group shows improvements, the intervention will be offered to the control group.

## Statistical analysis

**Quantitative analysis.** The data obtained as study variables will be coded, processed and analysed by the members of the research group at the end of the collection. SPSS5 will be used for statistical analysis. A confidence level of 95% and/or a statistical significance of $p < 0.05$ will be considered.

In the case of participants who have not completed the study due to dropout, have not fully completed treatment, or have even changed treatment, any data that may have been collected will be analysed for intent to treat. The minimum participation rate in the intervention in order to consider the results will be 80%.

To describe the sample, descriptive statistics will be performed, analysing the mean, standard deviation, minimum, maximum and percentages of the sociodemographic variables. In this way, it will be possible to describe the sample and observe the homogeneity between groups.

Next, a univariate and bivariate statistical analysis will be performed. First, it will be assessed whether or not the data follows a normal distribution for each of the variables. This analysis will determine whether to use parametric or non-parametric tests in the analysis of results based on the p value.

Equality of variance will then be analysed using the Levenne test, and finally, the analysis of the results will be performed in order to compare proportions and means. There will be a pre- and post-intervention analysis between and within groups, with a 3-month follow-up. We can also analyse possible correlations between variables.

**Qualitative analysis.** Qualitative analysis is a non-linear, interactive and dynamic process. It will consist of different parts following Colaizzi's 7-stage method [43], which is based on: 1 – Reading and rereading the transcript; 2 –Extracting significant statements that pertain to the phenomenon; 3 –Formulating meanings from significant statements; 4 –Aggregating formulated meanings into theme clusters and themes; 5 –Developing an exhaustive description of the phenomenon's essential structure or essence; 6 –Generating a description of the fundamental structure of the phenomenon; and 7 –Validating the findings of the study through participant feedback. The analysis will be performed using the program Atlas.ti 8.0.

# Discussion

There is little evidence to link GMI to PLP, and that which exists is very heterogeneous. The systematic review carried out previously for the design of the protocol shows that the three techniques that make up the GMI are effective in reducing PLP [44–48], but those studies that combine the three, that is, that use GMI as a whole, are more effective in reducing pain [16, 25]. Evidence suggests that neuroplastic changes for pain relief occur in response to internally generated manipulation through the intervention [14]. The GraMI protocol seeks the reorganisation of brain plasticity through the laterality recognition, motor imagery, motor work, sensory interaction with objects and functional activities, through observation and movements of the ipsilateral side to areas injured by amputation [49]. Interventions based on sensory stimulation training or active exercises are thought to restore the activation of the cortical representation of the amputated limb, leading to the activation of these zones and reducing phantom limb pain [50]. This protocol allows, through new technologies, to promote the active participation of the patient. In this way, the patient can carry out the intervention at home, enhancing the intensity of the treatment. In addition, new technologies, in this case mobile applications, encourage adherence and motivation to the intervention [51]. This protocol may help standardize the use of GMI in future studies and in clinical practice with this patient profile. One of the limitations of the study is that not all participants in the control group will receive the same treatment since it is a multicenter study in a community setting. The main line of treatment in the clinical setting for phantom limb pain is pharmacological. For this reason, one of the inclusion criteria is that all participants are receiving pharmacological treatment.

# Supporting information

**S1 File. SPIRIT 2013 checklist.**
(DOCX)

**S2 File. Research project submitted to the ethics committee (English).**
(DOCX)

**S3 File. Research project submitted to the ethics committee (Catalan).**
(DOCX)

**S4 File. Approved certificate of the ethics committee.**
(DOCX)

## Author Contributions

**Conceptualization:** Sandra Rierola-Fochs, Eduard Minobes-Molina.

**Data curation:** Sandra Rierola-Fochs.

**Investigation:** Jose Antonio Merchán-Baeza, Eduard Minobes-Molina.

**Methodology:** Sandra Rierola-Fochs, Jose Antonio Merchán-Baeza, Eduard Minobes-Molina.

**Supervision:** Sandra Rierola-Fochs, Jose Antonio Merchán-Baeza, Eduard Minobes-Molina.

**Validation:** Sandra Rierola-Fochs, Jose Antonio Merchán-Baeza, Eduard Minobes-Molina.

**Writing – original draft:** Sandra Rierola-Fochs, Jose Antonio Merchán-Baeza, Eduard Minobes-Molina.

**Writing – review & editing:** Sandra Rierola-Fochs, Jose Antonio Merchán-Baeza, Eduard Minobes-Molina.

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
