## [Decision Letter · Decision Letter 0]

27 Jun 2022

PONE-D-22-04167

EFFECTIVENESS OF GRADED MOTOR IMAGERY (GRAMI) PROTOCOL IN PHANTOM LIMB PAIN IN AMPUTED PATIENT: PROTOCOL OF A RANDOMIZED CLINICAL TRIAL

PLOS ONE

Dear Dr. Merchán-Baeza,

Thank you for submitting your manuscript to PLOS ONE. After careful consideration, we feel that it has merit but does not fully meet PLOS ONE’s publication criteria as it currently stands. Therefore, we invite you to submit a revised version of the manuscript that addresses the points raised during the review process.

We look forward to receiving your revised manuscript.

Kind regards,

Walid Kamal Abdelbasset, Ph.D.

Academic Editor

PLOS ONE

2. Please include a copy of Table 1 which you refer to in your text on page 6.

Reviewers' comments:

Reviewer's Responses to Questions

**Comments to the Author**

1. Does the manuscript provide a valid rationale for the proposed study, with clearly identified and justified research questions?

Reviewer #1: Yes

Reviewer #2: Yes

2. Is the protocol technically sound and planned in a manner that will lead to a meaningful outcome and allow testing the stated hypotheses?

Reviewer #1: Yes

Reviewer #2: Yes

3. Is the methodology feasible and described in sufficient detail to allow the work to be replicable?

Reviewer #1: Yes

Reviewer #2: Yes

4. Have the authors described where all data underlying the findings will be made available when the study is complete?

Reviewer #1: Yes

Reviewer #2: Yes

5. Is the manuscript presented in an intelligible fashion and written in standard English?

Reviewer #1: Yes

Reviewer #2: Yes

6. Review Comments to the Author

You may also provide optional suggestions and comments to authors that they might find helpful in planning their study.

Reviewer #1: Thank you for your effort, but please explain

When was the protocol started after amputation?

Define what was the limb included in the protocol, amputated or non amputated limb.

More information about the result and its better to add pictures and tables

More information about the control group

Reviewer #2: Title

1. Title needs to be modified please remove the abbreviation from the title

2- certain sections in the methods need references as mirrorr therapy

7. PLOS authors have the option to publish the peer review history of their article (what does this mean?). If published, this will include your full peer review and any attached files.

Reviewer #1: No

Reviewer #2: No

---

## [Author Response · Author response to Decision Letter 0]

8 Jul 2022

Walid Kamal Abdelbasset, Ph.D.

Academic Editor

PLOS ONE

We would like to thank the reviewers for their thoughtful and constructive comments. We have fully considered their suggestions and have incorporated them into the revised manuscript. Changes to the original manuscript are made in colour. We believe that the quality of our manuscript has improved as a result of the modifications. An itemized point-by-point response to the reviewers’ comments is presented below.

Reviewer 1:

Thank you for your effort, but please explain

When was the protocol started after amputation?

Thank you for your comment. The protocol is designed to be applied in the community setting; the participant can perform it autonomously at home under the supervision of the principal investigator. One inclusion criteria (see table 1) is that the participant must be discharged from the hospital. This criteria is established to not interfere with rehabilitation during hospital admission and so the participant is pharmacologically stable.

Define what was the limb included in the protocol, amputated or non amputated limb.

Thank you for your comment. After an amputation, there is an alteration in the laterality recognition due to neuronal malaplasticity. The inclusion criteria (see Table 1) is that the participant has amputated a unilateral limb since the mirror therapy technique could not be performed if there was a bilateral amputation. During the first laterality recognition technique, images of both sides of the body will appear through the application so that the participant works on recognizing bilateral laterality. In the next phase, the motor imagery, the image bank in the mobile application is configured so that 60% of images come from the amputated side and 40% from the healthy side, in this way we work on cerebral malplasticity, enhancing the affected side.

More information about the result and its better to add pictures and tables

Thanks for your input. Figures with the progression of images in the different phases of the protocol have been added to the manuscript to make it more visual. In addition, the expected benefits of this intervention have been added to the discussion on page 12.

More information about the control group

We thank the reviewer for their comments and have tried to further develop the control group intervention. One of the limitations of the study is that not all participants in the control group will receive the same treatment since it is a multicenter study in a community setting. The main line of treatment in the clinical setting for phantom limb pain is pharmacological. For this reason, one inclusion criteria (see Table 1) is that all participants receive pharmacological treatment. The participants in the control group will continue with the treatment they are undergoing at that time, or if they do not receive it, they will only receive the pharmacological treatment. At the same time, a follow-up diary will be given to the control group participants so that they write down all those interventions that influence the pain of the phantom limb.

This information has been added on page 9 in “control group intervention” and as a limitation of the study on page 13 at the end of the discussion.

Reviewer 2: Title

1. Title needs to be modified please remove the abbreviation from the title

Thank you for your comment. The title abbreviation has been removed

2- Certain sections in the methods need references as mirror therapy

We thank the reviewer for this observation. References have been added to the methodology section. On page 7 line 215, the reference to the intervention protocol has been added. On the same page, line 246, the reference to the KVIQ scale has been added to assess motor imagery capacity and the strategies to be considered in the event of alteration. On page 8, line 218, three references have been added in the mirror therapy section to argue the intention of movement of the affected limb during the technique.

---

## [Decision Letter · Decision Letter 1]

8 Aug 2022

EFFECTIVENESS OF GRADED MOTOR IMAGERY PROTOCOL IN PHANTOM LIMB PAIN IN AMPUTED PATIENT: PROTOCOL OF A RANDOMIZED CLINICAL TRIAL

PONE-D-22-04167R1

Dear Dr. Merchán-Baeza,

We’re pleased to inform you that your manuscript has been judged scientifically suitable for publication and will be formally accepted for publication once it meets all outstanding technical requirements.

Kind regards,

Walid Kamal Abdelbasset, Ph.D.

Academic Editor

PLOS ONE

Additional Editor Comments (optional):

Reviewers' comments:

Reviewer's Responses to Questions

**Comments to the Author**

1. Does the manuscript provide a valid rationale for the proposed study, with clearly identified and justified research questions?

Reviewer #1: Yes

2. Is the protocol technically sound and planned in a manner that will lead to a meaningful outcome and allow testing the stated hypotheses?

Reviewer #1: Yes

3. Is the methodology feasible and described in sufficient detail to allow the work to be replicable?

Reviewer #1: Yes

4. Have the authors described where all data underlying the findings will be made available when the study is complete?

Reviewer #1: Yes

5. Is the manuscript presented in an intelligible fashion and written in standard English?

Reviewer #1: Yes

6. Review Comments to the Author

You may also provide optional suggestions and comments to authors that they might find helpful in planning their study.

Reviewer #1: Thanks for your effort,

All the comments have been addressed

7. PLOS authors have the option to publish the peer review history of their article (what does this mean?). If published, this will include your full peer review and any attached files.

Reviewer #1: No

---

## [Editor Report · Acceptance letter]

16 Aug 2022

PONE-D-22-04167R1 

Effectiveness of Graded Motor Imagery protocol in phantom limb pain in amputed patient: protocol of a randomized clinical trial 

Dear Dr. Merchán-Baeza:

I'm pleased to inform you that your manuscript has been deemed suitable for publication in PLOS ONE. Congratulations! Your manuscript is now with our production department. 

Kind regards, 

on behalf of

Dr. Walid Kamal Abdelbasset 

Academic Editor

PLOS ONE